# The Antiglycoxidative Ability of Selected Phenolic Compounds—An In Vitro Study

**DOI:** 10.3390/molecules24152689

**Published:** 2019-07-24

**Authors:** Agnieszka Piwowar, Anna Rorbach-Dolata, Izabela Fecka

**Affiliations:** 1Department of Toxicology, Faculty of Pharmacy, Wroclaw Medical University, Borowska Str. 211, 50-556 Wrocław, Poland; 2Department of Pharmacognosy and Herbal Medicines, Faculty of Pharmacy, Wroclaw Medical University, Borowska Str. 211A, 50-556 Wrocław, Poland

**Keywords:** glycoxidation, AOPPs, AGEs, antiglycoxidative potential, phenolic compounds

## Abstract

Hyperglycemia and oxidative stress may be observed in different diseases as important factors connected with their development. They often occur simultaneously and are considered together as one process: Glycoxidation. This can influence the function or structure of many macromolecules, for example albumin, by changing their physiological properties. This disturbs the homeostasis of the organism, so the search for natural compounds able to inhibit the glycoxidation process is a current and important issue. The aim of this study was the examination of the antiglycoxidative capacity of 16 selected phenolic compounds, belonging to three phenolic groups, as potential therapeutic agents. Their antiglycoxidative ability, in two concentrations (2 and 20 µM), were examined by in vitro study. The inhibition of the formation of both glycoxidative products (advanced glycation end products (AGEs) and advanced oxidation protein products (AOPPs)) were assayed. Stronger antiglycoxidative action toward the formation of both AOPPs and AGEs was observed for homoprotocatechuic and ferulic acids in lower concentrations, as well as catechin, quercetin, and 8-*O*-methylurolithin A in higher concentrations. Homoprotocatechuic acid demonstrated the highest antiglycoxidative capacity in both examined concentrations and amongst all of them. A strong, significant correlation between the percentage of AOPPs and AGEs inhibition by compounds from all phenolic groups, in both examined concentrations, was observed. The obtained results give an insight into the antiglycoxidative potential of phenolic compounds and indicate homoprotocatechuic acid to be the most promising antiglycoxidative agent, but further biological and pharmacological studies are needed.

## 1. Introduction

Serum albumin is the most important multifunctional protein circulating in the blood. It has many important biochemical and physiological properties and plays a number of essential functions in the organism. Albumin binds a variety of endo- and exogenous molecules (e.g., hormones, fatty acids, metal ions, drugs) and displays enzymatic activity, as well as possessing antioxidative properties [1,2]. For this reason, any qualitative and/or quantitative changes in serum albumin, which may occur as the result of the actions of various factors, may result in adverse disturbances in its structure and function, leading to an imbalance in organism homeostasis [3,4]. Among these imbalances, hyperglycemia and oxidative stress are common in different diseases, for example, in diabetes mellitus (DM). Albumin is especially sensitive to such processes as glycation and oxidation, although especially interesting, and most important, is the coexistence of both of these states, known as the glycoxidation process, acting toward albumin modification [5,6]. These are the most important damaging agents relative to serum albumin and other macromolecules by their glycation and/or oxidation [3,5,7]. These processes in the organism are complex and multistage. They are chronic and often occur simultaneously, mutually intensifying their adverse actions. To emphasize their importance, they are usually considered together and specified as one: The glycoxidation process [6,8].

Originally only high-glucose concentration was burdened with adverse effects on macromolecules and tissues, especially in patients with DM, as a consequence of persistent long-term hyperglycemia and nonenzymatic reactions between aldehyde or ketone groups of reducing sugars with amino groups of proteins, such as the Maillard reaction [6,9,10]. An initial formation of unstable compounds occurs in this process (the so-called Schiff base), and after subsequent rearrangement of these structures, more stable forms (so-called Amadori products) are created, which are early glycation products. Finally, after many complex reactions (e.g., dehydration, cyclization, and condensation, as well as partial oxidation), irreversible, stable compounds, known as advanced glycation end products (AGEs), are formed [11,12,13]. The presence of free amino groups (e.g., 30-35 lysine residues) in the albumin molecule may be a target of the Maillard reaction [14]. There are many other specific or single products of glycation, such as glycated hemoglobin, pentosidine, carboxymethyllysine (CML), carboxyethyllysine (CEL), glyoxal (GO), methylglyoxal (MGO), and 3-deoxyglucosone (3DG) [12,13,15,16]. Oxidative stress appends are indicated as the other significant component of metabolic and clinical disturbances in diabetes. Initially, only changes in single parameters of oxidative/antioxidative balance were examined by radical scavenging assays: Ferric reducing ability (FRAP), 1,1-diphenyl-2-picrylhydrazyl radical scavenging activity (DPPH), oxygen radical absorbance capacity (ORAC), or thiol group levels [17,18,19]. But lately, attention has also been paid to the formation of more complex modified macromolecules, such as advanced oxidation protein products (AOPPs). AOPPs formation is significantly connected with increased production of free radicals and intensive chloramine derivative action on albumin, as well as subsequent complex modifications of its structure (e.g., degradation, cross-links and adduct formation) [20,21]. The most common results of glycoxidation are increased formation of both advanced glycation end products (AGEs) and advanced oxidation protein products (AOPPs). The participation and role of AGEs and AOPPs, especially in the development of DM disturbances, have been proven [11,22], but their role in other diseases, such as cardiovascular disease, renal and liver dysfunction, Alzheimer’s disease, cancer development, and the aging process, is also indicated [23,24,25].

Due to the growing epidemic of DM and other diseases of civilization [26,27], the significant participation of AGEs and AOPPs in their pathogenesis and development is underlined. Therefore, it is very important to identify potentially therapeutic compounds for prolonged usage to prevent serum albumin from glycoxidative modification. More and more evidence indicate that hypoglycemic agents and natural antioxidants, derived from the food and diet supplements, provide effective protection against the negative effects of hyperglycemia and oxidative stress [9,28,29], and in this way, may attenuate the adverse action of AGEs and AOPPs in the organism. For this reason, the simultaneous inhibition of protein glycation and oxidation is desirable, but most research concerns the ability to inhibit these processes separately (either glycation or oxidation) by various compounds. They refer for example to the inhibition of AGEs formation or single products of glycation (CML, CEL, CO, MGO, 3DG), as mentioned above [12,13,15,16]. The examination of oxidation processes, however, is mostly connected with the total antioxidant capacity or the oxidative/antioxidative balance [17,18,19].

In the light of the desired antiglycoxidative action, phenolic compounds, which constitute a large family of plant components, are a very interesting group. Additionally, they have numerous different activities and are commonly consumed with the diet. Their daily intake ranges from 100 mg to 2 g per day [30,31]. Some phenolic acids and their esters and depsides (rosmarinic acid, chlorogenic acid, ellagic acid, and other ellagitannin metabolites), as well as flavan derivatives (flavan-3-ols, flavonols, flavones, etc.), belong to this group. Both chlorogenic and rosmarinic acids occur naturally in various fruits, vegetables, plant spices, and beverages, as well as in some medicinal herbs. Simple phenolic acids such as caffeic, ferulic, homoprotocatechuic, protocatechuic, gallic, and others are the main metabolites of polyphenols, e.g., caffeoyl derivatives, tannins, anthocyanin, and flavonoid glycosides, after partial fragmentation by gut microbiota [32,33,34,35]. Plant phenolics exhibit many different activities, e.g., antimicrobial, anti-inflammatory, antineoplastic, vasoprotective, hepatoprotective, and atheroprotective, as well as antioxidative, which is generally the most well-known activity [36,37,38]. Some literature data also indicate the antiglycative properties of phenolic compounds [39,40,41]. However, to date, little is known about their ability to inhibit the glycoxidation process. To our knowledge, only single studies precisely concern the possibilities of inhibition of both AGEs and AOPPs formation [12,42,43]. Hence, there is still a need to search for different agents with antiglycoxidative properties and the ability to reduce both AGEs and AOPPs levels.

For that reason, the aim of our study was to evaluate and compare the antiglycoxidative capacity of 16 common plant phenolics or their potential metabolites by assessing their ability to inhibit the formation of AGEs and AOPPs in concentrations which are possible to obtain with the diet or supplementation. They represent three chemical groups of phenolic compounds: Phenolic acids and their esters (protocatechuic, homoprotocatechuic, caffeic, ferulic, rosmarinic, and chlorogenic acids); ellagic acid and other ellagitannin metabolites (urolithins and methylurolithins); and flavan derivatives (kaempferol, quercetin, myricetin, catechin). 

## 2. Results

All examined phenolic compounds used in the experiments were categorized into three groups, as shown in Table 1: Phenolic acids and their esters (group 1.); ellagic acid and other ellagitannin metabolites (group 2.); and flavan derivatives (group 3.). Additionally, substances used as well-known reference inhibitors of glycoxidation are given (group 4.). In this table their appropriate abbreviations, molecular weights, and chemical structures are also shown.

Figure 1, Figure 2 and Figure 3 illustrate the effectiveness of all examined compounds relative to inhibition of the glycoxidation process. The bars on the figures show the percentage of AOPPs and AGEs generated during glycoxidation in samples incubated with 2 and 20 µM of tested phenolics, as well as with known reference glycoxidation inhibitors (AA at 100 μM and AMG at 500 μM), in relation to the control (ctrl) where glycoxidation has been completed. All significant differences between these samples and ctrl (signed as described in Section 4.7) are also given. A 20% inhibition was accepted as the cut-off value for effective inhibition of glycoxidation.

All examined phenolic acids and their esters (group 1.), used in both concentrations, significantly influence the generation of AOPPs, but to varying degrees (Figure 1A). The inhibition of AOPPs creation by PA was similar for both its concentrations (~27%). FA and ChA act more strongly in higher concentrations. FA acts more effectively than ChA (28% vs. 17%), but there were no statistical differences between the used concentrations of these compounds. The remaining compounds from this group show stronger antiglycoxidative properties at lower concentrations. The most significant inhibitory effect was observed for 2 μM of DHPAA (35%), which was significant and about 1.3 times stronger than 20 μM. At lower concentrations, CA and its dimer RA have similarly strong inhibitory actions toward AOPPs generation (29% and 28%, respectively). This was significantly different from those observed for higher concentrations. We revealed that only DHPAA at 2 μM demonstrated a greater glycoxidation inhibitory effect than AA, a known reference inhibitor (35% vs. 30%). Furthermore, we noted that CA and RA at 2 μM, as well as FA at 20 μM and PA at both concentrations, demonstrated very similar anti-AOPPs potential compared with AA.

Analysis of the influence of phenolic acids and their esters on AGEs formation (Figure 1B) revealed that most of them act effectively at both used concentrations, but also to varying degrees. Contrary to the significant inhibition of AOPPs, PA inhibits AGEs formation weakly (~10%), and significant differences were observed only for its higher concentration. A comparable inhibition effect was detected for CA at 2 and 20 μM (17% and 18%). For DHPAA, FA and ChA, a stronger inhibitory action was observed for lower concentrations (29%, 20%, and 18%, respectively). Only for FA were significant differences revealed between used concentrations. RA in lower concentrations inhibited AGEs only slightly (~3%), but when used in higher concentrations, inhibited their formation significantly (by 19%). Additionally, a significant difference was observed between its used concentrations. In general, higher concentrations of phenolics from the examined group 1., except for RA, did not display any additional beneficial inhibitory action. Moreover, for none of the examined compounds did we observe a more inhibitory effect on glycoxidation in comparison to AMG (46%), a known reference inhibitor. Among the selected phenolic acids and their esters, DHPAA at 2 and 20 μM, as well as FA at 2 μM, demonstrated the greatest anti-AGE effect, but still less in comparison to AMG (~2 times).

The percentage of AOPPs inhibition by phenolics from group 2.—ellagic acid and other ellagitannin metabolites, like dibenzo-α-pyranone derivatives (urolithins and methylurolithins) at examined concentrations (Figure 2A), was mostly at a similar level to that mentioned above for phenolic acids and their esters. All differences between these compounds and ctrl were significant. EA and UA, as well as MUA and DMUC, inhibit AOPPs formation somewhat more strongly at 20 μM concentrations, in contrast to UB and UC, which act more effectively at 2 μM concentrations. The most effective antiglycoxidative action toward AOPPs formation was observed for EA (29%). UB inhibits AOPPs somewhat more strongly than UC (26% vs. 23%). The percentage of AOPPs inhibition reported for UA in both concentrations (26% and 22%) was higher than that of its methyl derivative - MUA (21% and 17%), at 2 and 20 μM, respectively. In the case of UC and its dimethyl derivative (DMUC), this tendency was similar only for its lower concentration (23% vs. 18%). At 20 μM, a stronger inhibitory action was revealed for DMUC (19% vs. 24%). Compared to AA, only EA at 20 μM demonstrated a similar anti-AOPPs effect (30% vs. 29%). 

Contrary to the inhibition of AOPPs formation, EA and derivatives of dibenzo-α-pyranone had a somewhat weaker influence on the inhibition of AGEs generation (Figure 2B). This ranged from 26% for MUA at 20 μM to a lack of inhibitory effect for DMUC at the same concentration. There were no significant differences for EA or UA or for MUA or DMUC when used in lower concentrations, or for UB and DMUC in higher concentrations. The percentage of AGEs inhibition by UA and UB at both concentrations was low, ranging from somewhat above 6% for UB at 20 μM, to below 10% for UA at 2 μM. The inhibitory action of UC and EA at 20 μM was slightly higher (13% and 16%). Unexpectedly, the percentage inhibition of AGEs generation by the methyl derivative of urolithin A (MUA) at 20 μM was meaningfully greater, but we did not observe an analogue inhibitory effect for DMUC. Generally, higher concentrations of phenolics from the examined group 2. displayed a slight beneficial inhibitory effect against the formation of AGEs. Significant differences between both used concentrations were observed only for EA and MUA. No compound belonging to group 2 acted as effectively as AMG as a reference inhibitor. Only MUA at 20 μM demonstrated relatively strong anti-AGEs potential, but still less in comparison to AMG (about one third). 

Among tested flavan derivatives (group 3.) with flavanol and flavonol structures (Figure 3A), most (excepting KAE) inhibit AOPPs formation, but to a different degree. A general tendency toward stronger inhibition of AOPPs generation with its higher concentration was observed, in the range from 8% for MYR, up to 26% for QUE, although significant differences between both used concentrations of these three compounds were observed. The highest inhibitory effect on AOPPs formation out of both tested concentrations was demonstrated by QUE. Moreover, CAT at 20 μM also shows a significant ability to inhibit AOPPs formation. Among the flavan derivatives, only QUE at 20 µM demonstrated a high anti-AOPPs effect, close to AA, a known reference inhibitor. 

As we can observe from Figure 3B, all tested flavan derivatives, used in both examined concentrations, influence the generation of AGEs, but to varying degrees. A similar trend to the stronger inhibition of AGEs formation at higher concentrations was observed for flavonols (KAE, QUE, MYR). However, KAE showed a rather weak (below 10%), and nonsignificant inhibitory effect. For MYR at 20 μM, the anti-AGEs effect was slightly higher, and significant (13%). The highest anti-AOPPs activity was detected for QUE at 20 µM concentration (28%). An opposite trend was observed for CAT–the lower concentration demonstrated somewhat greater inhibition effect than the higher concentration (24% vs. 18%). Significant differences between the ability to inhibit AGEs, in both used concentrations of QUE, CAT, and MYR were, however, observed. Among group 3., none of the examined compounds showed similar anti-AGEs potential to AMG.

The obtained data clearly show that only five phenolic compounds (DHPAA, FA, QUE, CAT, and MUA) demonstrate effective inhibition of the simultaneous formation of both products of the glycoxidation process–AOPPs and AGEs. Assuming 20% effective inhibition of the glycoxidation process as a cut-off point, we observed that the majority of the tested compounds more effectively inhibited the generation of AOPPs (as many as 13 out of 16) than AGEs (only five). 

Multiple regression analysis conducted between the percentage of inhibition of AOPPs and AGEs formed during the glycoxidation process in the presence of tested phenolics at both used concentrations revealed a strong, significant correlation in all examined groups of compounds. In group 1.–phenolic acids and their esters–the correlation coefficient (*r*) values were 0.946 and 0.937 for 2 and 20 μM, respectively. In the ellagic acid and other ellagitannin metabolites (group 2.), the *r* values were 0.998 and 0.960. For group 3., independent of concentrations used, *r* was similarly high: 0.993 and 0.995.

## 3. Discussion

In the present work, we conducted screening tests of the antiglycoxidative ability of 16 common plant phenolics or their potential metabolites belonging to three groups: 1. Phenolic acids and their esters, 2. ellagic acids and other ellagitannin metabolites, and 3. flavan derivatives (Table 1). We chose these phenolic compounds because of their widespread presence in plants and many beneficial properties, as well as their possible participation in the reduction of oxidative stress and lowering hyperglycemia [38,44,45]. Their metabolism and biological availability vary and depend on their chemical structure [46,47]. After consumption, phenolic compounds are liberated from plant tissues as primary compounds, and after ingestion by enzymes, they are absorbed in the intestine and metabolized in the typical pathway of detoxification of xenobiotics. They can be absorbed as primary compounds, aglycones, or low-molecular compounds, or as both primary and secondary forms simultaneously [31,44,48,49]. EA and urolithins are the main low-molecular catabolites of ellagitannins, such as punicalagin, agrimoniin and sanguiins. Flavan derivatives occur in plasma after the consumption of different flavonoids, mostly flavonol glycosides and flavanol gallates. DHPAA and PA are products of C-ring fission of flavan-derived compounds such as anthocyanins, flavonols, flavanols, and dimeric procyanidins [31,45,46,50]. 

We examined the antiglycoxidative ability of these 16 selected phenolics in two concentrations, 2 and 20 μM, established on the basis of literature data, and possible to obtain in the blood plasma from food (in the daily diet) and/or supplementation [31,32]. As an experimental model of glycoxidation, we used BSA in the physiological concentration occurring in the blood plasma, as well as glucose and chloramine-T in appropriate concentrations, as recommended in the literature for fast and efficient conduction of this process [3,51]. Additionally, L-ascorbic acid and aminoguanidine (Table 1, group 4.) were used as reference inhibitors with well-known antioxidative and antiglycative properties [52,53].

We decided to apply the measurement of concentrations of both final products of the glycoxidation process, i.e., AOPPs and AGEs, which are very helpful and sufficient for the purposes of screening tests, since they are strictly related to monitoring the intensity of glycoxidation. We revealed that most of the tested phenolics, in the used concentrations, inhibit the formation of both these products, but to varying degrees. A general tendency toward stronger inhibition of AOPPs than AGEs formation was observed, because, as we showed among selected compounds, 13 were more effective in reducing AOPPs than AGEs (only five compounds), when we apply a 20% inhibitory level as a limit of satisfactory inhibition of the glycoxidation process. Moreover, the strength of inhibition was also greater with regard to AOPPs (up to 35%) when compared to AGEs generation (up to 29%).

Some differences in antiglycoxidative effect were observed when we considered particular groups (1–3) of compounds. The most effective seems to be phenolics belonging to group 1.–simple phenolic acids and their esters–all of them, in both used concentrations, significantly inhibited AOPPs formation, but only DHPAA and FA also significantly reduced AGEs formation. Slightly higher anti-AOPPs and anti-AGEs potential was reported for a phenolic acid with a catechol ring (3,4-dihydroxyphenyl) and 2-carbon chain than phenolic acids with 3-carbon or 1-carbon chain (DHPAA vs. PA, CA). Substitution of one OH functional group with methyl did not considerably change these properties of phenolic acid in lower concentrations (CA vs. FA), but differences in higher concentration were evident. There were no higher inhibitory abilities observed when the dimer was applied (CA vs. RA). The esters have comparable or somewhat lower antiglycoxidative action than free phenolic acids (ChA and RA vs. CA). The above-mentioned observations make phenolic acids the most interesting and promising, especially DHPAA and FA, taking into account the fact that they express higher anti-AOPPs and anti-AGEs activities in low concentrations. Moreover, CA, RA, and PA are interesting as potential inhibitors of AOPPs and moderate inhibitors of AGEs formation. 

Phenolic compounds belonging to group 2.–ellagic acid and other ellagitannin metabolites–also present substantial anti-AOPPs action (all reduced AOPPs by more than 17%). Only a slight difference in the antiglycoxidative potential of dibenzo-α-pyrone derivatives with different OH/OMe groups in the molecule was observed. The most effective was EA at 20 μM, characterized by the presence of two lactones and four OH groups (from two catechol rings) in the molecule. Monolactones (urolithins and methylurolithins) with a minor number of phenol groups were less active. An interesting observation concerns MUA at 20 μM, which showed significant antiglycoxidative ability, effectively decreasing both AOPPs and AGEs formation. Comparing the percentage of AGEs inhibition recorded for UA and its methyl derivative MUA, a greater effect of inhibition was revealed for MUA. This may show a possible role of methoxyl in anti-AGEs action. 

Interestingly, compounds from group 3–flavan derivatives–have proven to be rather stronger AGEs than AOPPs inhibitors. Two of these four tested compounds–QUE and CAT–effectively reduced AGEs formation. As shown, derivatives of flavan with a 3′4′-dihydoxyphenyl ring were more active than those with 3’,4’,5’-trihydroxyphenyl or 4’-hydroxyphenyl (QUE and CAT vs. MYR and KAE). The presence or absence of carbonyl at C-4 had an insignificant inhibitory effect (QUE vs. CAT). For flavonols (but not for flavan-3-ol), inhibitory action toward AGEs generation increased with the concentration. These compounds with catechol rings also inhibit AOPPs formation, so this makes them the next potential antiglycoxidative agents. Aglycones with three OH groups (MYR) and with only one OH group in the phenyl ring (KAE) remained less active or inactive. These observations are consistent with the results obtained from randomized, double-blind, placebo-controlled, and crossover clinical trials from 2018, which investigated the inhibitory effects of individual flavonoids on AGEs and reactive α-dicarbonyl compounds. In a human intervention study, Van den Eynde et al. [16] found a significant reduction in plasma MGO concentrations by QUE supplementation (ingested as quercetin-3-*O*-glucoside, 160 mg/day), but not by epicatechin (100 mg/day). 

The multiple regression analysis made between percentage inhibition of AOPPs and AGEs generated in the presence of the examined phenolic compounds showed a strong correlation between these parameters (*r* around 0.9), which was mostly independent of the concentration of the tested phenolics. This does not provide a definite answer regarding the character of the inhibition of glycoxidation by these compounds, but points to a strong link between both glycation and oxidation processes in the in vitro study. Therefore, the application of some of the phenolic compounds we examined becomes very promising. Taking into account their structure (presence of OH, OMe, carboxyl and carbonyl functional groups) indicates that phenolic acids and their esters can be potent inhibitors of glycoxidation, but the strongest correlation was observed for flavan derivatives, as well as ellagic acid and other ellagitannin metabolites. Phenolics as inhibitors of AGEs and AOPPs formation have attracted great interest among researchers, and suggested mechanisms of their action are currently widely discussed. However, none of these authors have examined both anti-AGEs and anti-AOPPs effects simultaneously. The results of the aforementioned studies show the possibility of their antiglycative action at various steps of AGEs generation and indicate a connection between the chemical structure of these molecules and their activity [13,41]. The following structural elements are suggested to be the most important for flavonoids: Hydroxylation of A and B rings, methylation, glycosylation, and hydrogenation of the C2/C3 double bond. The above observations are in agreement with our results for flavan derivatives (QUE>CAT>MYR>KAE at 20 μM). In their in vitro study, Verzelloni and coworkers [9] found that ellagitannin metabolites, including urolithins A and B, significantly decreased albumin glycation (37–44%) at concentrations of 1 and 2 μM. The antiglycative activity of EA, urolithins A and B, as well as punicalagin, was also confirmed by Liu et al. [54], who revealed that both punicalagin and EA were the most potent inhibitors of glycation compared to guanidine as a reference inhibitor. Previous experiments with usage of ellagitannins and ellagic acid showed that hydroxylation increased anti-AGEs activities, but methylation decreased them [13,40,41]. We perceived a similar effect for ellagic acid and urolithins (EA > UC > UA > UB at 20 μM), but not for all the tested methylurolithins (UA < MUA and UC > DMUC at 20 μM). 

Fairly well-understood and known hypoglycemic and/or antiglycative properties toward AGEs formation are demonstrated, for example, by various teas, grapeseed and soybean extracts, and also by individual flavonoids, phenolic acids, and others. Additionally, their antioxidant activities are also shown [39,40,55,56]. RA is especially emphasized, not only because of its ability to prevent preformed AGEs-related albumin crosslinks, but also because of its promising anti-diabetic and anti-aging properties [57]. Jayanthy et al. [58] observed that oral administration of RA to streptozotocin-induced diabetic rats triggered the decrease of blood glucose concentration and the level of HbA1c. Sompong et al. [59] revealed that FA reduced the levels of AGEs, CML, and protein carbonyl content in glucose-, fructose-, and ribose-glycated BSA models. Interestingly, the decrease of AGEs by FA correlated with the levels of protein glycation and oxidative damage of BSA. Lin et al. [60] also reported that PA provided antiglycative, anti-inflammatory, and anti-coagulatory protection against diabetic complications in vivo. In the animal model, EA attenuated diabetic retinopathy in rats through the inhibition of AGEs formation under hyperglycemic conditions [61].

Reduction of oxidative stress, trapping free radicals, chelating transition-metal ions, and decomposition of intermediate products (mainly reactive carbonyl species referred to as RCS) are indicated as indirect inhibitory mechanisms, weakening and preventing the oxidative damage of macromolecules, among others, by phenolics [62]. Serum albumin modifications in hyperglycemic conditions are mainly related to the reaction of RCS (α-dicarbonyls, e.g., glucosone, 3-DG, GO, MGO) with amino groups of the protein, which can be intensified by oxidative stress, and their concentrations increased in DM [16,63]. Usui et al. [10] suggest that α-dicarbonyls are generated from glucose via non-oxidative 3-deoxyglucosone formation, and oxidative glucosone formation in non-enzymatic glucose degradation. Moreover, the presence of transition-metal ions (even traces) accelerate the formation of RCS, and subsequently AOPPs and AGEs. RCS intermediates act as potent cross-linkers, and they can be considered the key stage of protein glycoxidation [64,65], which will be interesting for further study of these phenolic compounds.

Therefore, we can assume two ways of antiglycoxidative action of tested phenolic compounds: *i.* Protection of glucose against oxidation and nonenzymatic degradation (and lowering of the RCS level) or *ii.* protection of serum albumin from modification by RCS. It is likely that because phenolics can act as antioxidants (through free radical scavenging and transition-metal ions’ chelating activities), they can efficiently trap reactive α-dicarbonyl intermediates or bind with serum albumin by hydrogen bonds and/or through hydrophobic interactions, preventing glycoxidative modification, and all these mechanisms can coexist or complement each other. It is also suggested that especially compounds with phenolic and carbonyl functional groups may interact with the same serum albumin sites as RCS. 

Published data indicate the binding affinity of flavonoids and phenolic acids to serum albumin, which, in this way, may enhance their protective efficiency against glycoxidation [66,67,68,69,70]. These phenolics spontaneously interacted with serum albumin, and their binding affinity was relatively lower for phenolic acids and depsides and stronger for flavonoids [68,71,72,73,74]. However, the structural differences of tested compounds (the number and position of free phenols, esterification of carboxyl and presence of carbonyl groups) remarkably affected the binding process. Phenolics with tri-substituted (e.g. pyrogallol) or di-substituted (catechol) structures exhibited stronger binding affinities than mono-substituted derivatives [68,71]. It was found that methoxyl substituting the phenol group decreased affinity to BSA [68,73]. Moreover, it was shown that hydrogen bonds and hydrophobic interaction play a major role in phenolic binding to serum albumin, and esters (e.g., 3-O-gallates of flavan-3-ols or ChA) have higher binding affinity than non-esterified compounds (e.g., CAT, CA) [71,73,75]. 

The inhibition of AGEs formation by trapping MGO and/or GO was also indicated for flavonoids (e.g., KAE, QUE, epicatechin), DHPAA, and simple phenols (e.g., pyrogallol, catechol) [63,76,77,78]. Among these compounds, 1,2,3-trihydroxybenzene, flavonols and flavones showed the highest scavenging activity by trapping MGO. In another study, EA showed MGO scavenging properties comparable with AMG, whereas urolithins A and B were weaker [9,54]. Shao et al. [79] showed that position C-5 of pyrogallol, as well as C-6 and C-8 of the flavonoid A ring, are the major active sites for trapping RCS. Mono- and di-MGO adducts are formed by flavonoids with the OH groups at C-7 and C-5. The double bond between C-2/C-3 on the C (heterocyclic) ring could facilitate trapping efficacy, but the structure of B (phenyl) rings is not engaged. The revised hemiacetal structures of the MGO-quercetin adducts were elucidated by Bhuiyan at al. [80]. 

As has been shown, phenolic compounds react with ROS, and the reaction rate of each phenolic depends on its ability to form a stable radical. Compounds with a 1,2-dihydroxybenzene, 1,4-dihydroxybenzene, or 1,2,3-trihydroxybenzene ring are easier oxidized to a phenoxyl semiquinone radical which can be stabilized by the second oxygen atom. Monophenols, 1,3-diphenols, and substituted phenols (especially methoxy derivatives) are not as readily oxidized, because they do not produce stabilized semiquinone radicals. Both catechol and pyrogallol groups seem to be primary reacting structures with ROS, and they form *o*-semiquinone radicals. *O*-semiquinone radicals can be further oxidized to *o*-quinones and through tautomerization deliver *o*-quinone methides. Quinones and *o*-quinone methides can bind with proteins [81]. Bhuiyan at al. [80] propose an inhibitory mechanism of QUE against AGEs formation including a chelation effect, trapping MGO and ROS, which leads to oxidative degradation of the QUE-di-MGO adduct to PA and other stable fragments. Herein, we confirmed that PA possesses anti-AOPPs and anti-AGEs abilities at low concentrations. It seems that the final products of RCS deactivation by flavonoids with 3′4′-dihydoxyphenyl may retain positive antiglyoxidative activity. On the other hand, Leopoldini et al. [82] and Liu et al. [83] indicated that among the potential sites of chelation present in flavonoids, the oxygen atoms belonging to 5-hydroxypyran-4-one and 3-hydroxypyran-4-one, rather than the 3’4’-dihydroxyphenyl in the B ring, were preferred. Therefore, we can conclude that phenolics with vicinal hydroxyl groups show stronger anti-AOPPs and anti-AGEs properties, and the 5,7,3’,4’-tetrahydroxyflavonoid structure is vital to the antiglycoxidative activity of tested compounds. Figure 4 shows possible chelating and trapping sites of quercetin and exemplary structures of reaction products.

More and more evidence indicate that antioxidants and antihyperglycemic agents in the diet or supplements provide effective protection against the negative action of hyperglycemia and excessive free radical generation [9,28], and in this way, may attenuate the adverse effects of AOPPs and AGEs in the organism. So, the obtained results from our in vitro study seem to be promising and suggest that phenolic compounds exhibit antiglycoxidative activity through inhibition of the formation of both glycoxidation products–AOPPs and AGEs. This means that in the case of diabetic conditions demonstrated by chronic supraphysiological concentrations of glucose and coexisting oxidative stress, it is advisable to enrich the diet with plant products rich in phenolic acid esters, flavan derivatives, and ellagitannins. 

## 4. Material and Methods

### 4.1. Phenolic Compounds and Other Chemicals 

Aminoguanidine (purity ≥98%); Bovine Serum Albumin (degree of purity ≥92%); chloramine-T (purity ≥98%); D-(+)-glucose (purity ≥99%); L-ascorbic acid (purity ≥99%); catechin and myricetin (purity of both ≥95%), derived from Sigma Aldrich, St. Louis, MO, USA. Phosphate buffered saline (PBS) prepared from appropriate amounts of NaCl, KCl, Na_2_HPO_4_, and KH_2_PO_4_ (reagents grade), and potassium iodide (reagents grade), were purchased from POCH, Gliwice, Poland. Caffeic and rosmarinic acids (purity of both ≥98%), and chlorogenic acid (purity ≥95%), derived from Fluka Chemica, Munich, Germany; acetic acid (reagent grade), from Merck, Darmstadt, Germany. Protocatechuic acid, ferulic acid, and 3,4-dihydroxyphenylacetic acid, (purity of all these ≥99%); kaempferol and quercetin (purity ≥98% and ≥95%, respectively), all derived from Extrasynthese, Genay, France; ellagic acid (purity ≥95%) was purchased from Koch-Light Laboratories LTD, Haverhill, Suffolk, England, UK. Urolithin A, urolithin B, urolithin C, 8-*O*-methylurolithin A, and 8,9-*O*-dimethylurolithin C were synthesized according to Bialonska et al. [28], the degree of purity of all of them ≥95% (designated by HPLC-PDA, column: Hypersil GOLD, 5 µm, 250 × 4.6 mm; Thermo Scientific, Loughborough, UK).

### 4.2. Preparation of Phenolic and Reference Compounds 

In the experiments, all the examined phenolics were used in two working concentrations: 2 µM and 20 µM. All the stock solutions were prepared in the concentration of 200 µM in PBS or DMSO (according to the manufacturer’s instructions). The tested mixtures (4 mL) were obtained by appropriate dilution with PBS at 1:10 (*v/v*). A blank control containing all reagents (without the addition of examined compounds) was made for each series of tests. The effect of DMSO on the glycoxidation process was also checked at the concentration used (10%, *v/v*) to dissolve the examined compounds and no effect was observed. For each series of experiments, a positive control (ctrl) was made. In this sample, albumin was incubated only with the glycating and oxidizing agents (glucose and chloramine-T, respectively) and the glycoxidation process was completed. In the (ctrl) sample the effectiveness of the glycoxidation process was assumed to be 100%. In addition, to control the effective inhibition of the glycoxidation process with use of aminoguanidine (AMG) and L-ascorbic acid (AA) as known reference inhibitors (Ref.), antiglycative and antioxidative agents were used. They were applied in concentrations 0.5 mM and 0.1 mM, respectively, which are those most commonly used in scientific literature data [84,85]. Even though these agents are not in concentrations similar to physiological conditions or to examined natural compounds, this allows for satisfactory quantities of a product to be obtained in a shorter time, and the results obtained in this experimental in vitro model permit preliminary screening to identify the most active antiglycoxidative compounds, as was also mentioned by other researchers [12,39].

### 4.3. Model of Albumin Glycoxidation and Antyglycoxidation Action 

For the conduction of glycoxidation, BSA solution in PBS, at a concentration of 40 mg/mL, was incubated at 37 °C (LaboPlay, series SWB, Poland) for 14 days with 30 mM glucose solution in phosphate buffered saline (pH 7.4), and in the presence or absence of analyzed phenolic compounds at final concentrations of 2 µM and 20 µM. Then chloramine-T (28 mM) was added to all samples and they were incubated for a period of 1 h at 37 °C. Sodium azide (1 mM) as a preservative was used in the experiments [9,86]. Subsequently, all samples were dialyzed and frozen until the time of measurement of glycoxidation products (AOPPs and AGEs).

### 4.4. Dialysis

In order to remove the excess of unbound substances (analyzed compounds, glucose, chloramine-T), dialysis in phosphate buffered saline (pH 7.4) was performed with a 100:1 buffer to sample volume ratio at 4 °C. All samples were moved to the dialysis membrane (10 kDa MWCO, Spectra/Por, Spectrum Laboratories, Inc., Phoenix, AZ, USA) and subjected to three buffer changes over a period of 24 h of dialysis as follows: First buffer change after 2 h, the second one after 4 h, and the last one after 24 h from the beginning of the dialysis procedure. Then all the samples were frozen at −80 °C until the time of measurement.

### 4.5. Fluorometric Measurement of AGEs 

After all samples had been thawed, they were centrifuged for 10 s at 6000 rpm. The level of AGEs was measured according to the method of Münch et al. [87], based on the measurement of characteristics for the fluorescence of the compounds at two wavelengths. The fluorescence was measured on a Perkin Elmer LS 50B spectrometer (USA) at the excitation maximum of 370 nm and emission maximum of 440 nm (apertures of 10 and 20 mm, respectively), taking into account the appropriate dilution (500×). The concentration of AGEs was determined according to the formula: AGEs [AFU] = (F × D) × 1000/10, where AFU is the arbitrary fluorescence units, F is the fluorescence intensity, D is the dilution, 1000 is the cuvette volume (µL), and 10 is the sample volume (µL).

### 4.6. Spectrophotometric Measurement of AOPPs 

The concentrations of formed AOPPs were measured according to the Witko-Sarsat et al. [21] method. All samples had been thawed before the conduction of experiments, and they were centrifuged for 10 s at 6000 rpm. The samples were diluted appropriately (100×) in PBS, followed by the addition of 50 µL of acetic acid. Within 10 s, 100 µL of 1.16 M potassium iodide was added to each sample and the measurements were made immediately. The absorbance was read at 340 nm in a Stat-Fax 1904 spectrophotometer (Thermo Fisher Scientific, Bellport, NY, USA) against a blank sample (containing only PBS, potassium iodide and acetic acid). The chloramine-T standard curve absorbance at 340 nm was linear (*r*^2^ = 0.9992) within the range of 0 to 100 µM. The AOPPs concentration was determined according to the formula: AOPPs [µM] = 1/B × A × D × 1000, where B is the slope of the standard curve, A is absorbance, D is the dilution, and 1000 is the cuvette volume (µL).

### 4.7. Data Analysis

The percentage values of the obtained results were calculated in relation to the positive control (ctrl), in which the glycoxidation process had reached 100% completion according to the formula: X% = $\bar{x}$/(ctrl) × 100%, where $\bar{x}$ is the average value of all measurements for created AGEs and AOPPs during the glycoxidation process with the addition of selected compounds. The average values for each compound were calculated as the arithmetic mean values from two series of three repetitions of each sample including standard deviations. The differences between samples and K(+), as well as between each other individual sample (used at 2 µM and 20 µM concentrations), were analyzed using Student’s test in Statistica PL version 13, and the following statistical significance values, with appropriate symbols (given in all figures), were established: (*) *p* < 0.001, (•) *p* < 0.01, (#) *p* < 0.05. The percentage inhibition of the formed glycoxidation products was calculated as the difference of (ctrl) sample (%) and the average value of AOPPs and AGEs formation (%) for individual examined compounds. Additionally, multiple regression analysis was performed to examine the relationship between the inhibition of AOPPs and AGEs created during the glycoxidation process in the presence of the tested phenolic compounds.

## 5. Conclusions

The phenolic compounds examined and selected by us may be considered useful as antiglycoxidative agents. Five of them–DHPAA and FA, CAT, QUE, as well as MUA–demonstrated strong anti-AOPPs and anti-AGEs properties during the in vitro glycoxidation process. This indicates their complementary potential in circumstances with simultaneous glycation and oxidation. Importantly, DHPAA significantly inhibits the generation of both AOPPs and AGEs at the micromolar concentration range. As our research is preliminary and screening in nature, there is, therefore, an urgent need to fully explore these promising results and conduct further research geared toward identifying and exhaustively evaluating these best antiglycoxidative phenolic compounds, with more emphasis on their pharmacological and toxicological profile.

## Figures and Tables

**Figure 1 molecules-24-02689-f001:**
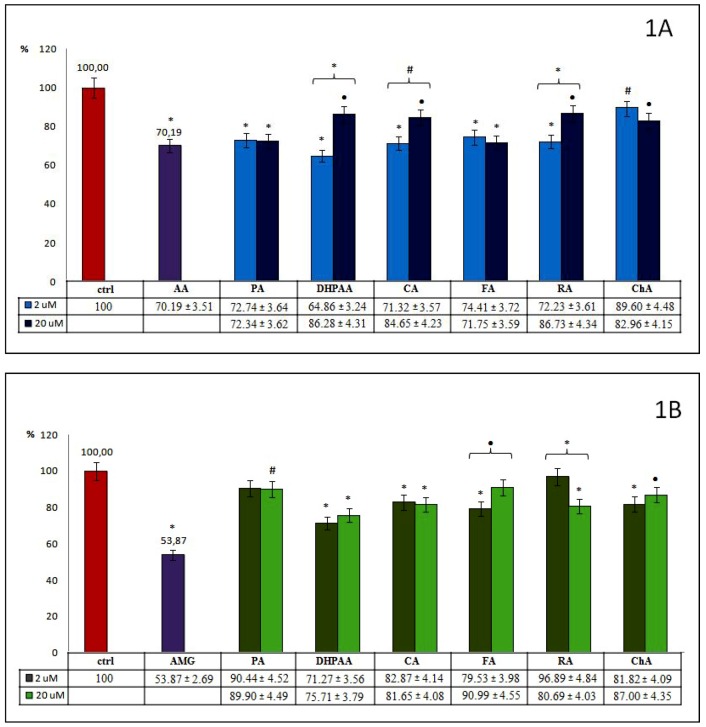
Percentage of advanced oxidation protein products (AOPPs) (**1A**) and advanced glycation end products (AGEs) (**1B**) formation during the glycoxidation process in samples incubated with 2 and 20 µM of tested phenolic acids and their esters (group 1.). Significant differences in comparison to (ctrl), as well as between both used concentrations, were expressed as: (*) *p* < 0.001, (•) *p* < 0.01, (#) *p* < 0.05.

**Figure 2 molecules-24-02689-f002:**
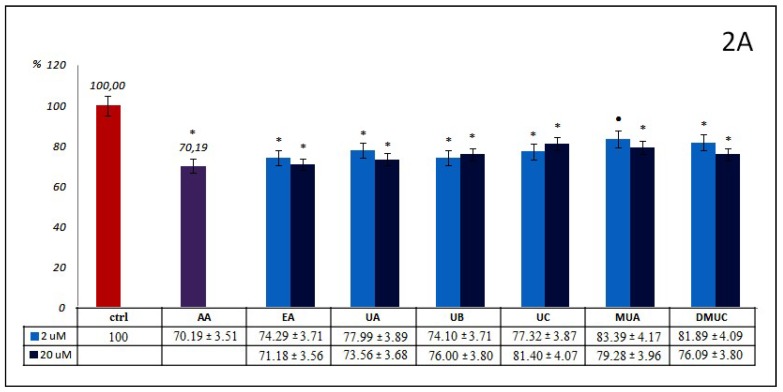
Percentage of AOPPs (**2A**) and AGEs (**2B**) formation during the glycoxidation process in samples incubated with 2 and 20 µM of tested ellagic acids and other ellagitannins metabolites (group 2.). Significant differences in comparison to (ctrl), as well as between both used concentrations were expressed as: (*) *p* < 0.001, (•) *p* < 0.01, (#) *p* < 0.05.

**Figure 3 molecules-24-02689-f003:**
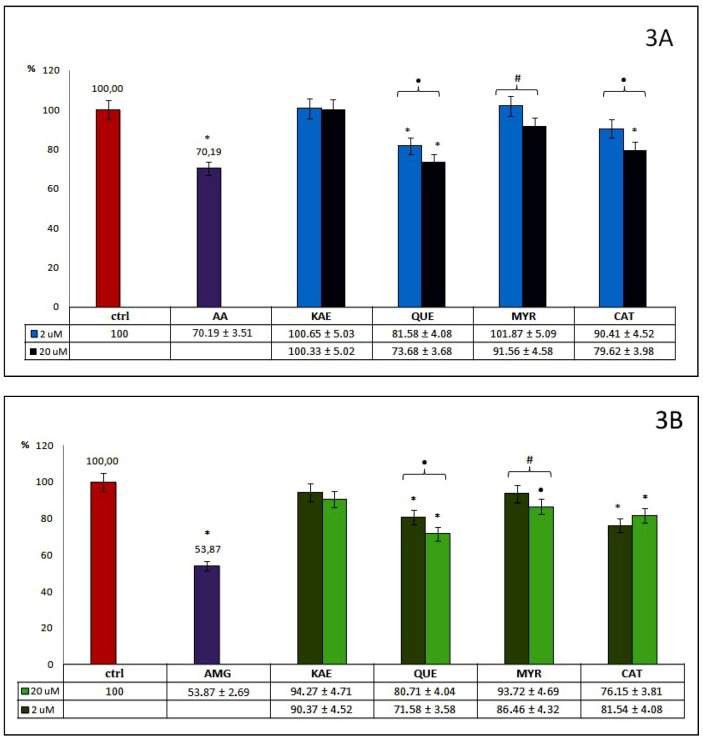
Percentage of AOPPs (**3A**) and AGEs (**3B**) formation during the glycoxidation process in samples incubated with 2 and 20 µM of tested flavan derivatives (group 3.). Significant differences in comparison to (ctrl), as well as between both used concentrations, were expressed as: (*) *p* < 0.001, (•) *p* < 0.01, (#) *p* < 0.05.

**Figure 4 molecules-24-02689-f004:**
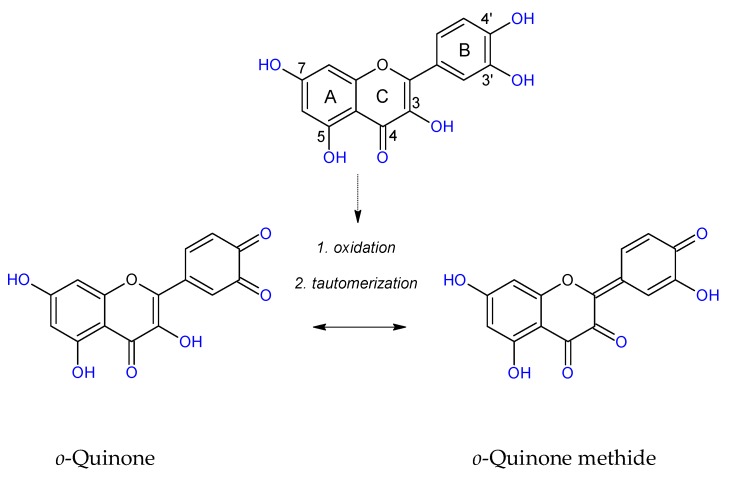
Possible chelating and trapping sites of quercetin, and exemplary structures of reaction products.

**Table 1 molecules-24-02689-t001:** Phenolic compounds and reference inhibitors of both glycation and oxidation processes.

Abbreviation	Compound	MW [g/mol]	Structure
Phenolic acids and their esters (group 1.)			
PA	Protocatechuic acid (3,4-dihydroxybenzoic acid)	154.12	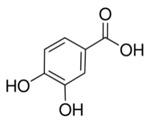
DHPAA	Homoprotocatechuic acid (3,4-dihydroxyphenylacetic acid)	168.15	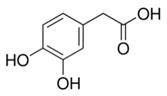
CA	Caffeic acid (3,4-hydroxycinnamic acid)	180.16	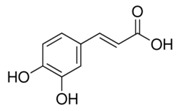
FA	Ferulic acid (4-hydroxy-3-methoxycinnamic acid)	194.18	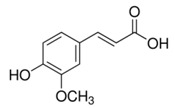
RA	Rosmarinic acid	360.31	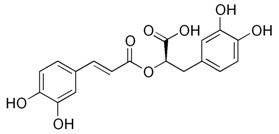
ChA	Chlorogenic acid	354.31	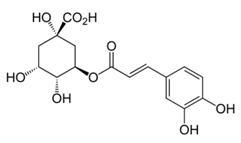
Ellagic acids and other ellagitannin metabolites (group 2.)			
EA	Ellagic acid	302.20	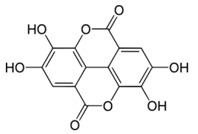
UA	Urolithin A	228.21	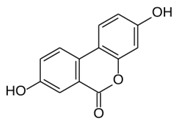
UB	Urolithin B	212.21	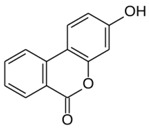
UC	Urolithin C	244.21	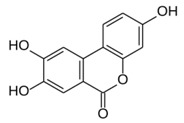
MUA	8-*O*-Methylurolithin A	242.23	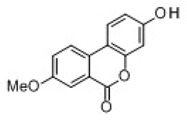
DMUC	8,9-*O*-Dimethylurolithin C	272.26	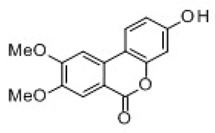
Flavan derivatives (group 3.)			
KAT	Kaempferol	286.24	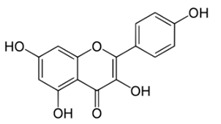
QUE	Quercetin	302.24	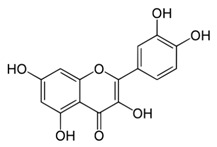
MYR	Myricetin	318.24	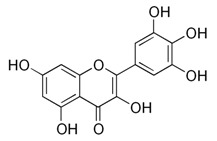
CAT	Catechin	290.26	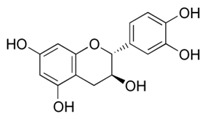
Reference inhibitors of glycation and oxidation (group 4.)			
AMG	Aminoguanidine	74.09	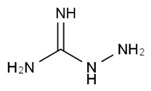
AA	L-Ascorbic acid	176.12	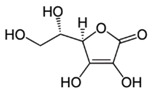

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
