# Peer review of "The Antiglycoxidative Ability of Selected Phenolic Compounds—An In Vitro Study"

_molecules, 2019, doi:10.3390/molecules24152689_

Reviewer 1 Report

The submitted paper describes the evaluation of antiglycoxidative ability of phenolic compounds based on the inhibitory effect on the formation of advanced glycation end products (AGEs) and advanced oxidation protein products (AOPPs). Although the findings described in the submitted paper may be meaningful, it should be difficult to accept for the publication in its present form due to following concerns.

(1)   The descriptions in the Discussion should be shortened and summarized. Since the authors described the characteristics of phenolic compounds with citations from previously reported works similar to a review article, the descriptions, it was obscure which discussion was based on the experimental results obtained by the authors. Therefore, it was difficult to evaluate the significance and advantage of the submitted paper. The discussion according to previous works should be suppressed to a necessary minimum.

(2)   The authors confirmed the statistical difference between phenol incubated groups and reference.  However, the statistically significant differences between 2 µM phenol incubated groups and 20 µM incubated groups were not evaluated (Figure 1-3). Since the submitted study largely discussed about the influence of the difference of tested concentrations on the antiglycoxidative ability, the confirmation of the statistical difference between 2 and 20 µM phenol should be necessary.   

(3)   Figure 4A and 4B could be omitted from the manuscript because the significance of these figures was only to give an example of effective anti-glycoxidation compounds. It could be revealed by only the descriptions in the manuscript.

(4)   The single letter abbreviation should be used essentially for amino acids. Therefore, the abbreviations used for group 3 should be revised (for example, Kaepfrol ->KAE, Quercetin-> QUE). In addition, the authors did not use the abbreviation for the tested compound (e.g. line 284-285 of page 11).

Author Response

REWIVER 1

 (1)   The descriptions in the Discussion should be shortened and summarized. Since the authors described the characteristics of phenolic compounds with citations from previously reported works similar to a review article, the descriptions, it was obscure which discussion was based on the experimental results obtained by the authors. Therefore, it was difficult to evaluate the significance and advantage of the submitted paper. The discussion according to previous works should be suppressed to a necessary minimum.

In answer to (1): We have shortened the Discussion section and added the Conclusions at the end of this paragraph. We tried to make this section simpler, but we also wanted to show and explain our hypothesis of the potential mechanism of action of these tested phenolics, which in our opinion seems to be important and interesting.

(2)   The authors confirmed the statistical difference between phenols incubated groups and reference.  However, the statistically significant differences between 2 µM phenol incubated groups and 20 µM incubated groups were not evaluated (Figure 1-3). Since the submitted study largely discussed about the influence of the difference of tested concentrations on the antiglycoxidative ability, the confirmation of the statistical difference between 2 and 20 µM phenols should be necessary.   

 In answer to (2): We added information about all statistical differences observed between the studied concentrations of each compounds (2 mM compared to 20 mM) on the figures 1-3.  If such differences existed, appropriate symbols (the same as earlier used) were applied to make these differences visible on the figures. In addition, we added information about these differences in the Results section.

(3)   Figure 4A and 4B could be omitted from the manuscript because the significance of these figures was only to give an example of effective anti-glycoxidation compounds. It could be revealed by only the descriptions in the manuscript.

In answer to (3): We wanted to better illustrate the antiglycoxidative properties of tested compounds but now we have removed Figure 4 from the manuscript, and the information about the percentage of antiglycoxidative ability of these compounds are given in the text of manuscript.

(4)   The single letter abbreviation should be used essentially for amino acids. Therefore, the abbreviations used for group 3 should be revised (for example, Kaempfrol ->KAE, Quercetin-> QUE). In addition, the authors did not use the abbreviation for the tested compound (e.g. line 284-285 of page 11).

In answer to (4): According to that commentary, we have replaced the one-letter abbreviations of tested compounds with three-letter abbreviations. Currently it is: K is KAE, Q is QUE, M is MYR, and C is CAT. In addition, we have introduce these abbreviations in the all text.

Reviewer 2 Report

Manuscript ID: molecules-546048

The object of this manuscript entitled "Antiglycoxidative ability of selected phenolic

compounds – an in vitro study”, is the study of in vitro antiglycant and antioxidant activities of several natural phenols.

The aim is not new, several papers on this matter are been already published by several authors. There are only few new progresses in comparison to the matter already known. Furthermore, many critical points are present in this research.

Major points:

The introduction is not specific, reporting only information on general role of glycation and ROS in DM. The  topic must be better written.  How did the authors pick the reported compounds? What was the reason? Also the abstract should be improved. The reference treatment (ref. in the manuscript) usually it is the control which is the 100% of the studied activity.

Table 1. The 8-O-Methylurolithin A and 8-O-Methylurolithin C structures were not reported. They should be add.

The abbreviation of Kaempferol (K), etc. were done with only the first letter; this may generate easily mistakes in the reader. I suggest to use always at least two letters. In general, in the manuscript it is hard follow the results description. For example, the sentence: “The inhibition of AOPPs creation by PA was very similar for both concentrations (~27%), whereas FA and ChA show somewhat stronger inhibition at higher concentration, but FA acts more effectively than ChA (28% vs. 17%).” appears without meaning. Authors should improve the text.

Data reported below the Figures are without the standard deviations; Why?

Why did Authors assume a cut-off point of 20% inhibition of the glycoxidation process? What is the experimental meaning? The cut-off of 20% is a low activity which appears as absence of activity.

Figure 4 is cryptic and overall unclear. To report each compound at different concentrations, in diverse position on the y-axis gives the figure difficult to understand.

The discussion is very long and have to be shorter, strictly related to the experimental results.

Minor points:

            Table 1. Reformat the caption in one more concise. E.g., “Phenolic compounds and reference inhibitors of both the glycation and the oxidation processes.”

 â€śFlavan-derived aglycones”: flavan-3-ol ?

Further, “Effective inhibitors of glycation and oxidation (group 4.)” change this in “Reference inhibitors”

MUA and MUC are without their respective structures.

The abbreviations of each compound's name should be longer than one only letter

There are many typographical mistakes; the text should be carefully checked.

 Author Response

REVIEWER 2

Major points: 

(1) The introduction is not specific, reporting only information on general role of glycation and ROS in DM. The topic must be better written.  How did the authors pick the reported compounds? What was the reason? Also the abstract should be improved. The reference treatment (ref. in the manuscript) usually it is the control which is the 100% of the studied activity.

In answer to (1): The Introduction section has been reworded and made more succinct. Therefore, we have added more detailed information about steps of glycation and oxidation processes, and appropriate references.

Referring to the pick of reported compounds: Phenolic compounds are a broad class of plant secondary metabolites. The most common compounds are flavonoid glycosides, anthocyanins, esters of caffeic acid and flavan-3-ols, as well as tannins, and they are found in herbal teas, spices, vegetables and fruits. These primary phenolics are digested after consumption and metabolized to simpler compounds like phenolic acids, urolithins and aglycones with the flavan structure. Only low-molecular phenolics (with molecular mass bellow 500 Da) are absorbed and detected in plasma and urine. For that reason we have decided to examine this selected 16 phenols from 3 groups: phenolic acids and their esters (group 1.); ellagic acid and other ellagitannin metabolites (group 2.); and flavan derivatives (group 3.).  It was also important for us that these compounds may come from diets or supplements and that they could probably help to inhibit of glycoxidative process in the human organism.

The Abstract has been checked and carefully reworded.

In accordance to Reviewer suggestion, we have changed abbreviation used to the samples where glycoxidation process gone completely (100%). It was earlier Ref., but currently it is signed as K(+). Additionally, the group 4. (aminoguanidine and ascorbic acid) we have described as “Reference inhibitors of glycation and oxidation”, and signed as Ref. , instead of “well known inhibitors of glycation and oxidation”.

(2) Table 1. The 8-O-methylurolithin A and 8-O-methylurolithin C structures were not reported. They should be adding.

In answer to (2): We improved the visibility of these compounds, and currently they are well visible.

(3) The abbreviation of Kaempferol (K), etc. were done with only the first letter; this may generate easily mistakes in the reader. I suggest to use always at least two letters. In general, in the manuscript it is hard follow the results description. For example, the sentence: “The inhibition of AOPPs creation by PA was very similar for both concentrations (~27%), whereas FA and ChA show somewhat stronger inhibition at higher concentration, but FA acts more effectively than ChA (28% vs. 17%).” appears without meaning. Authors should improve the text.

In answer to (3): We have changed all one-letter symbols of these tested compounds to three-letter abbreviations. Currently it is: K is KAE, Q is QUE, M is MYR, and C is CAT. In addition, we have introduced these abbreviations into the all text.

We have improved the description of the results, so it is easier to read. We have wanted to show the differences in the inhibitory activity of these various compounds belonging to the same group.

(4) Data reported below the Figures are without the standard deviations; Why?

In answer to 4): We added of SD values below the figures description. 

(5) Why did Authors assume a cut-off point of 20% inhibition of the glycoxidation process? What is the experimental meaning? The cut-off of 20% is a low activity which appears as absence of activity.

In answer to (5): We have examined phenolics in micromolar concentrations possible to obtain in the human plasma by supplying with diet or in the form of supplements. However, micromolar concentrations are lower than those which have been used most often in the other in vitro studies. Some researchers who used millimolar concentrations have observed a very effective inhibition of glycation or oxidation, but similar high concentrations are impossible to achieve in the organism. Probably they are also not safe. Thus, the 20% cut-off point of inhibition of glycoxidation may reflect the process in the human body. The 20% level of inhibition in the organism seems to be sufficient, effective and significant, so we have decided to apply this cut-off point.

(6) Figure 4 is cryptic and overall unclear. To report each compound at different concentrations, in diverse position on the y-axis gives the figure difficult to understand.

In answer to 6): We have removed figure 4 from our manuscript, and appropriate data are presented in the text, in the Results and partially in the Discussion.

(7) The discussion is very long and has to be shorter, strictly related to the experimental results.

 In answer to (7): We have shortened this section and tried to link it closely to our results. But we would also like to show our hypothesis about the possible way of action of these studied compounds, because we believe that this is an interesting aspect and is likely to be interesting for readers.

Minor points:

1.  Table 1. Reformat the caption in one more concise. E.g., “Phenolic compounds and reference inhibitors of the both the glycation and the oxidation processes.”

In answer to 1: We have changed the caption of Table 1 and added the appropriate description suggested by Reviewer.

2.  â€śFlavan-derived aglycones”: flavan-3-ol ?

In answer to 2: We have changed “Flavan-derived aglycones” on “Flavan derivatives” in the all body of manuscript. Additionally, we would like to explain that accordance to the UPAC recommendation flavan derivatives are compounds derived from 3,4-dihydro-2-phenyl-2H-1-benzopyran (e.g. flavanol, flavanones, flavanonols, flavonols, etc.). Consequently, CAT, KAE, QUE and MYR are flavan derivatives.

3. Further, “Effective inhibitors of glycation and oxidation (group 4.)” change this in “Reference inhibitors”

In answer to 3: We have changed the name of group 4. in the all body of manuscript, Aminoguanidine and ascorbic acid are currently described as reference inhibitors, assigned as Ref.

4. MUA and MUC are without their respective structures.

In answer to 4: We have improved the visibility of these compounds, and currently they are well visible.

5. The abbreviations of each compound's name should be longer than one only letter.

In answer to 5: We have changed of all one-letter symbols of tested compounds to three-letter abbreviations. Currently, it is: K is KAE, Q is QUE, M is MYR, and C is CAT.

6. There are many typographical mistakes; the text should be carefully checked.

This manuscript has been checked by native speaker.

Round  2

Reviewer 1 Report

The authors revised the manuscript appropriately according to the comments, and therefore the revised manuscript could be acceptable for the publication in its present form.

Author Response

REWIVER 1

(1).The authors revised the manuscript appropriately according to the comments, and therefore the revised manuscript could be acceptable for the publication in its present form.

In answer to (1):Thank you very much for your good opinion about our manuscript.

Reviewer 2 Report

There are still some mistakes of English. Some sentences are not clear to me. I've inserted the suggestions in the pdf of your manuscript. The Results are a little hard to read.

Author Response

REWIVER 2

(1).There are still some mistakes of English. Some sentences are not clear to me. I've inserted the suggestions in the pdf of your manuscript. The Results are a little hard to read.

In answer to (1):Thank you very much for your help  and comments. All the indicated remarks we included in the manuscript, and all these changes are marked in green in the text.